# Brax - A Differentiable Physics Engine for Large Scale Rigid Body Simulation

**C. Daniel Freeman**
Google Research

**Erik Frey**
Google Research

**Anton Raichuk**
Google Research

**Sertan Girgin**
Google Research

**Igor Mordatch**
Google Research

**Olivier Bachem**
Google Research

## Abstract

We present Brax, an open source library for **r**igid **b**ody simulation with a focus on performance and parallelism on accelerators, written in JAX. We present results on a suite of tasks inspired by the existing reinforcement learning literature, but remade in our engine. Additionally, we provide reimplementations of PPO, SAC, ES, and direct policy optimization in JAX that compile alongside our environments, allowing the learning algorithm and the environment processing to occur on the same device, and to scale seamlessly on accelerators. Finally, we include notebooks that facilitate training of performant policies on common OpenAI Gym MuJoCo-like tasks in minutes.

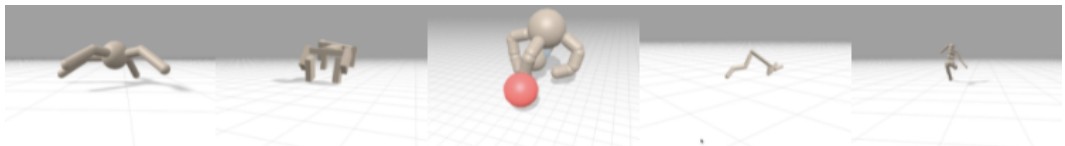

Figure 1: The suite of examples environments included in the initial release of Brax. From left to right: ant, fetch, grasp, halfcheetah, and humanoid.

## 1 Summary of Contributions

Brax trains locomotion and dexterous manipulation policies in seconds to minutes using just one modern accelerator. Brax achieves this by making extensive use of auto-vectorization, device-parallelism, just-in-time compilation, and auto-differentiation primitives of the JAX[1] library. In doing so, it unlocks simulation of simple rigidbody physics systems in thousands of independent environments across hundreds of connected accelerators. For an individual accelerator, Brax reaches millions of simulation steps per second on environments like OpenAI Gym's MuJoCo Ant[2]. See Sec. 6 for more details, or our Colab[3] to train a policy interactively.

The structure of the paper is as follows: we first provide motivation for our engine in Sec. 2. In Sec. 3, we describe the architecture of Brax, starting from the low level physics primitives, how they interact, and how they can be extended for practitioners interested in physics based simulation. In Sec. 4, we review our ProtoBuf environment specification, and detail how it can be used to construct rich physically simulated tasks, including the suite of tasks bundled in this initial release. In Sec. 5, we tour some of the reinforcement learning algorithms bundled with Brax. In Sec. 6, we catalog scaling behavior of Brax on accelerators, performance comparisons between Brax and MuJoCo on OpenAI Gym-style learning problems, and report Brax's engine fidelity on a benchmarking task.

35th Conference on Neural Information Processing Systems (NeurIPS 2021) Track on Datasets and Benchmarks.

In Sec. 7, we demonstrate how environment differentiability can be used to dramatically improve algorithm sample complexity on a simple task. Finally, in Sec. 8, we discuss the limitations and possible extensions of our engine.

## 2   Motivation

The reinforcement learning community has made significant progress on the study and control of physically simulated environments over the past several years. This progress stems from the confluence of strong algorithmic techniques [4–9] with accessible simulation software [10–14]. On the algorithmic side, model-free optimization techniques like proximal policy optimization (PPO)[6] and soft actor critic methods (SAC)[5] have exploded in popularity and can easily solve many of the "hard" control problems of the previous decade. On the simulation side, practitioners have the choice of a variety of engine backends to power their study of simulated environments, including MuJoCo[10], pybullet[15], and physX, among many others, many of which are differentiable[16–22, 14].

While these engines and algorithms are quite powerful, and have provided the firmament of algorithmic innovation for many years, they do not come without drawbacks. Reinforcement learning, as it is practiced, remains prohibitively expensive and slow for many use cases due to its high sample complexity: environments with only hundreds of dimensions of state space require millions to billions of simulation steps during RL exploration. As environments increasingly require interactive physics calculations as part of the environment step, this problem will only grow worse[23–25].

While some progress has been made to lower this sample complexity using off-policy algorithms[8, 26–28], RL systems instead frequently address sample complexity by scaling out the environment simulation to massive distributed systems. These distributed simulation platforms yield impressive RL results at nearly interactive timescales[29–32], but their hardware and power costs make them inaccessible to most researchers.

The design of the simulation engine contributes to this inaccessibility problem in three ways:

First, most simulation engines in use today run on CPU, while the RL algorithm runs on GPU or TPU, in another process or another machine. Latency due to data marshalling and network traffic across machines becomes the dominant factor in the time it takes to run an RL experiment.

Second, most simulation engines are black boxes: they do not offer a gradient for the sampled environment state, which makes them suitable only for model-free RL approaches. This lack of differentiability forces the researcher to use slower, less efficient optimization methods.

Finally, most simulation engines are black boxes in another way: they are either closed source, or built on an entirely different technical stack than the reinforcement learning algorithms. This lack of introspectability not only harms productivity by limiting rapid iteration and debugging, but it prevents researchers from understanding the relationship between the environment's state and action space, which is often critical to guiding new RL research.

We submit Brax as a proposed solution to all three problems at once. Brax puts a physics engine and RL optimizer together on the same GPU/TPU chip, improving the speed/cost of RL training by 100-1000x. It is differentiable, opening the door to new optimization techniques. And it's an open source library that is packaged to run in Colabs, so that anyone can do RL research for free.

## 3   Using Brax: The core physics loop

Brax simulates physical interactions in maximal coordinates[33], where every independent entity in a scene that can freely move is tracked separately. This data—position, rotational orientation, velocity, and angular velocity—is typically the *only* data that changes dynamically in the course of a simulation. All other dynamical relationships, like joints, actuators, collisions, and integration steps are then built as transformations on this fundamental state data. This is codified in the data primitive *qp*, implemented as a flax[34] dataclass, and named whimsically after the canonical coordinates $q$ (position) and $p$ (momentum) that it tracks. These *qp* datastructures thus store position, quaternion, velocity, and angular velocity data. To make vectorization easy, *qp*s have leading batch dimensions for the number of parallel scenes as well as the number of bodies in a scene. For example the shape

of the position data for 4 parallel scenes with 10 bodies per scene would be $[4, 10, 3]$—4 scenes, 10 bodies, and 3 spatial dimensions.

```python
def pseudo_physics_step(qp, action, dt):
    qp = kinematic_integrator.apply(qp, dt)
    for jo in joints:
        dp_j += jo.apply(qp)
    for ac in actuators:
        dp_a += ac.apply(qp, action)
    for co in colliders:
        dp_c += co.apply(qp)
    qp = potential_integrator.apply(qp, dp_j + dp_a, dt)
    qp = collision_integrator.apply(qp, dp_c)
    return qp
```

**Algorithm 1:** Pseudocode for the structure of a physics step in Brax. Impulsive updates ($dp_i$) are collected in parallel for each type of joint, actuator, and collider. These updates depend only on the dynamical state data for the system—the *qp*—which stores position, quaternion, velocity, and angular velocity data for all parts in the system. Integrator transformations then advance time by applying these impulses to the *qp*.

A physically simulated object generally includes extra data, thus we bundle other information— masses, inertias, dimensions of objects, etc.—in abstractions associated with particular *qp*s. These abstractions are *joints*, *actuators*, and *colliders*. Each of these abstractions has its own *apply* method that calculates an impulsive update ($\delta p$) based on the current state of the *qp* data, and optionally a user-defined action in the case of actuators:

$$\delta p = \text{Brax\_abstraction.apply}(qp)$$

To complete the physics step, Brax then sums up all of the differential, impulsive updates to *qp* data in the course of a single short timestep, and transform the system state via a symplectic Euler update (extensions to higher order integrators are straightforward, but see 8 for more details). These integrator functions return an update to the state data directly:

$$qp' = \text{Brax\_integrator.apply}(qp)$$

Throughout, we parallelize wherever possible, across actuators, joints, colliders, and even entire simulation scenes. See Alg. 1 for pseudocode for the structure of this loop, or [35] for the code of the loop. An overarching *system* class handles the coordination and bookkeeping of all of these updates and physical metadata. This class also provides a way to perform a single simulation step via the *step* function, which advances the state data by $\delta t$:

$$qp_{t+\delta t} = \text{Brax\_system.step}(qp_t,\ actions,\ \delta t)$$

where *actions* are any torques or target angles needed by any actuators in the system.

Modifying or extending this control flow is as simple as implementing a new Brax transformation that conforms to these structures, and then appropriately inserting this transformation in the physics step function.

In order to better visualize and explore Brax's core physics loop, please see our basics Colab [36].

## 4   Using Brax: Creating and evaluating environments

Brax provides an additional abstraction layer for interacting with physically simulated scenes. In Sec. 4.1, we describe the ProtoBuf specification for defining Brax systems—i.e., the lowest level data that describes any physics constraints in a system. Next, In Sec. 4.2, we motivate the *env* class, which allows practitioners to construct gym-like decision problems on top of Brax *systems*. Finally, we discuss the environments that have been prepackaged with Brax.

### 4.1 System specification

Our ProtoBuf text specification allows a user to define all of the *bodies* in a scene, how they are connected to each other via *joints*, as well as any *actuators* or *colliders* between objects, pairwise. For any tree of *bodies* connected by *joints*, Brax's *system* class will automatically determine the position and rotation of the *qp* that places each body in a valid joint configuration through the system.default_qp method.

Reminiscent of, e.g., MuJoCo's xml-based system definition, users can define systems in text, or they can define systems programmatically. We provide an example configuration that defines a joint between a parent and child body in Appendix A, both in the pure text form, and the programmatic form. Similar configuration files define every system in the Brax repo within each respective environment file, e.g. [37]. See our introductory Colab notebooks for an interactive tour of both of these apis.

### 4.2 Gym-like environments

For sequential decision problems, we must track extra metadata beyond what is necessary for a physics update. We provide an *env* class for handling book-keeping of any initializing, resetting, observing, acting, or reward function defining required to fully specify a sequential decision problem. We also provide a wrapper around this class to interface directly with it as an OpenAI gym-style interface.

To illustrate the versatility of Brax as an engine, we include and solve several example environments in our initial release: MuJoCo-likes (Ant, Humanoid, Halfcheetah), Grasp (a dexterous manipulation environment), and Fetch (a goal-based locomotion environment). See Table 1 for the dimension data for these environments.

| Env Name | Obs Dim | Act Dim | Type |
|----------|---------|---------|------|
| Halfcheetah | 25 | 7 | continuous |
| Ant | 87 | 8 | continuous |
| Humanoid | 299 | 17 | continuous |
| Grasp | 139 | 19 | continuous |
| Fetch | 101 | 10 | continuous |

Table 1: Observation and action space data for the environments included in Brax.

#### 4.2.1 MuJoCo Gym-Likes

The reinforcement learning and control communities have used the OpenAI Gym MuJoCo tasks as benchmarks for developing algorithms for the past several years. While these tasks are well-understood, and essentially solved, we provide our own fairly faithful reconstructions of three of these environments as a baseline point of comparison to help ground practitioner expectations. Owing to subtle engine differences, these environments are not perfectly identical to the MuJoCo scenes on which they are based, and we call out major differences in Appendix E.

#### 4.2.2 Grasp

Dexterous manipulation tasks have exploded in popularity as algorithmic and hardware advances have enabled robots to solve more complicated problems. *Grasp* is a simple pick-and-place environment, where a 4-fingered claw hand must pick up and move a ball to a target location. We include this environment primarily as a proof-of-concept to demonstrate that the contact physics of our engine are sufficient to support nontrivial manipulation tasks. For a representative sample trajectoy of a successful policy, see Fig. 7 in Appendix B.

#### 4.2.3 Fetch

We performed extensive experimentation on a variety of goal-directed locomotion tasks. *Fetch* represents a generally stable environment definition that is able to train a variety of morphologies to locomote within 50 million environment frames. For this release, we include a toy, boxy dog-like quadruped morphology as the base body, but it is straightforward to modify this scene for new body morphologies.

# 5 Using Brax: Solving locomotion and manipulation problems

To train performant policies on the environments included in this release and interactively evaluate them, see our training Colab[3].

## 5.1 Learning Algorithms Bundled with Brax

Brax includes several common reinforcement learning algorithms that have been implemented to leverage the parallelism and just-in-time-compilation capabilities of JAX. These algorithms are:

- Proximal Policy Optimization (PPO) [6]
- Soft Actor Critic (SAC) [4]
- Evolution Strategy (ES) [32]
- Analytic Policy Gradient (APG)

Each algorithm is unique in some respects. PPO is an on-policy RL algorithm, SAC is off-policy, ES is a black-box optimization algorithm, and APG exploits differentiability of the rewards provided by the environment. This breadth of algorithmic coverage demonstrates the flexibility of Brax, as well as its potential to accelerate research and reduce costs. For this work, we focus our experimental analysis on PPO and SAC (see, e.g., Sec 6), and defer analysis of ES and APG to future work.

### 5.1.1 Proximal Policy Optimization (PPO)

In order to capture all benefits of a JAX based batched environment that could run on an accelerator(s) we built a custom implementation of PPO. In particular the environment data (rollouts) are generated on an accelerator and subsequently processed there by an SGD optimizer. There's no need for this data to ever leave the accelerator nor is there any need for context switches between various processes. The whole training loop (env rollouts + SGD updates) happens within a single non-interrupted jitted function.

The training proceeds as follows:

- the batch is split evenly between every available accelerator core and environment rollouts are collected
- normalization statistics are computed based on this batch, stats are synced between all cores and then observations are normalized
- each accelerator core splits the batch into an appropriate number of mini batches for which gradient updates are computed, synced between all cores, and then applied synchronously

The performance/throughput of the algorithm heavily depends on the hyperparameters (e.g. batch size, number of minibatches, number of optimization epochs). We noticed that for the best hyperparameters, our implementation of PPO is efficient enough that the primary bottleneck comes from the environment(e.g., 75% time goes to running the env for Ant), even though the environment itself is quite fast.

### 5.1.2 Soft Actor Critic (SAC)

Unlike PPO, SAC uses a replay buffer to sample batches from. In order to use the whole potential of Brax we implemented a custom SAC with a replay buffer living completely on an accelerator. This allowed the whole training procedure to be compiled into a single jitted function and run without any interruptions. The training roughly proceeds as follows:

- each available accelerator core runs the environment for a few steps and adds this data to an individual per-core replay buffer
- normalization statistics are computed based on the newly generated data, stats are synced between all cores
- several SGD updates are performed, where each accelerator core samples its part of a batch from its own replay buffer, computes gradient updates, and synchronizes the final update with other cores

SAC is much more sample efficient than PPO, thus we observed that the training throughput now becomes bottlenecked by SGD updates (12% for running the env, 10% for working with replay buffer, 78% for SGD updates). Because of the poor scaling of SGD updates to multiple cores, using more than 1 accelerator core was providing marginal benefit, so the most cost efficient setup was achieved with a single accelerator core.

### 5.1.3 Evolution Strategy (ES)

To implement ES we followed the same paradigm as for PPO/SAC: we ran everything on an accelerator without any interruptions, keeping all processing contained within the accelerator.

The training proceeds as follows:

- a lead accelerator generates policy parameters perturbations
- policy parameters perturbations are split evenly between all available accelerator cores for evaluation
- the lead computes gradients based on evaluation scores and updates the policy

The algorithm spends > 99% of running time evaluating environment steps.

### 5.1.4 Analytic Policy Gradient (APG)

As a proof of concept of how to leverage the differentiablity of our engine, we provide a APG implementation. Training is significantly simpler than the previous algorithms:

- compile a function that takes a gradient of the loss through a short trajectory
- perform gradient descent with this function

After compiling the gradient update, this algorithm spends the majority of the remaining time evaluating the gradient function. This algorithm is less mature than the previous three, and does not currently produce locomotive gaits, and instead seems prone to being trapped in local minima on the environments we provide, with the exception of the Reacher task discussed in section 7 where it succeeds. Differentiating through long trajectories is an active area of research[38, 21, 18] and is known to be difficult to optimize[39, 40], thus we defer more advanced differentiable algorithms to future releases.

### 5.2 Training Performance

As part of our release, we include performant hyperparameters for all of our environments. These hyperparameters typically solve their environment with a standard accelerator in seconds to minutes. For exhaustive listings of our hyperparameter experiments see our **repo**[41]. For plots of performance of the best 20 hyperparameter settings for each environment for exhaustive hyperparameter sweeps over SAC and PPO, see Appendix D.

## 6 Performance Benchmarking

### 6.1 Parallelizing over Accelerators

By leveraging JAX's vectorization and device parallelism primitives, we can easily scale Brax up to hundreds of millions of steps per second of performance by distributing environment computation within and across accelerators. Fig. 2 depicts these scaling curves for the suite of environments included in this release on a particular fast, modern accelerator cluster (4x2 topology of TPUv3), as well as the performance scaling on the Ant environment for a variety of accelerators and TPU topologies. For reference, Colab TPU instances currently provide limited free usage of 2x2 TPUv2 accelerators.

### 6.2 Engine Comparisons

A perfectly apples to apples comparison between engines is difficult, primarily because the main way to parallelize the most widely used engines is either by custom multithreading harnesses over CPU, or

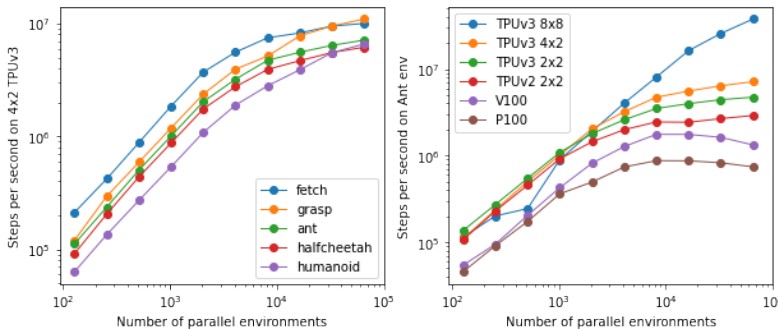

Figure 2: (left) Scaling of the effective environment steps per second for each environment in this release on a 4x2 TPU v3. (right) Scaling of the effective environment steps per second for several accelerators on the Ant environment. Error bars are not visible at this scale. These environments have between 8 (halfcheetah) and 17 (grasp) rigid bodies per scene, thus performance typically starts saturating beyond about 20,000 total rigid bodies per accelerator. See App. F for more details about the scaling of performance with number of rigid bodies in a scene.

by distributed aggregation of headless workers with attached accelerators—typically bespoke setups not available to most practitioners. Thus, it probably isn't fair to compare Brax's Ant environment compiled to and running on a TPUv3 8x8 accelerator (~hundreds of millions of steps per second) to the typical use case of a practitioner running the OpenAI gym MuJoCo-ant on a single threaded machine (~thousands of steps per second). While we include Brax results from deployment on large clusters of TPUs, we emphasize that Brax performance on a single 1x1 TPUv2 is significantly better than what the vast majority of practitioners have, until now, been able to achieve at dramatically reduced cost.

To make this performance gap clear, we first consider a qualitative comparison of training speed for the Ant environment with Brax's PPO implementation over a variety of architectures. We compare this to a traditional setup, with a standard implementation of PPO[28]—i.e., not compiled nor optimized for parallelism, visualized in Fig. 3. Note that Brax reaches performant locomotion in ten seconds or so, whereas the standard PPO implementation takes close to half an hour.

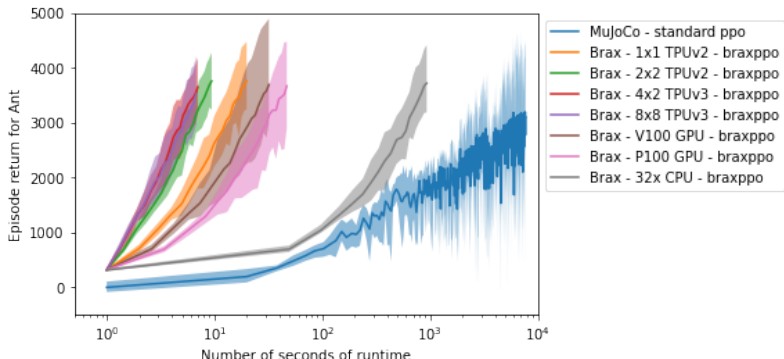

Figure 3: Qualitative comparisons of training curves for Brax's compiled and optimized PPO implementation versus a standard PPO implementation[28]. Note the x-axis is log-wallclock-time in seconds. All curves with "brax" labels are Brax's version of Ant, whereas the MuJoCo curve is MuJoCo-Ant-v2. Both implementations of ppo were evaluated for 10 million environment steps. Shaded region indicates lowest and highest performing seeds over 5 replicas, and solid line indicates mean. See App. C for hyperparameters used.

Next, to verify that Brax's versions of MuJoCo's environments are qualitatively similar to MuJoCo's environments, we depict training curves for a standard implementation of SAC on our environments side-by-side with training curves for MuJoCo's versions. Qualitatively, for a fixed set of SAC hyperparameters, Brax environments achieve similar reward in a similar number of environment steps

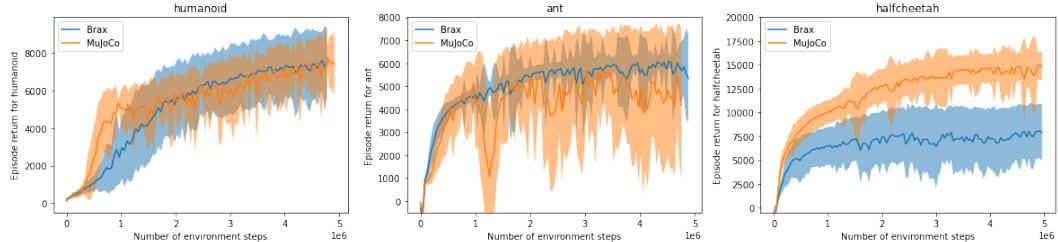

Figure 4: Qualitative comparisons of training curve trajectories in MuJoCo and Brax. (left) Training curves for MuJoCo-Humanoid-v2 and brax-humanoid, (middle) MuJoCo-Ant-v2 and brax-ant, and (right) MuJoCo-HalfCheetah-v2 and brax-halfcheetah. All environments were evaluated with the same standard implementation of SAC[28] with environments evaluated on CPU and learning on a 2x2 TPUv2—i.e., *not* Brax's accelerator-optimized implementation. Solid lines indicate average performance, envelopes are variance over random seeds. See App. C for hyperparameters used. See Appendix E for a short discussion of the gap in performance for halfcheetah.

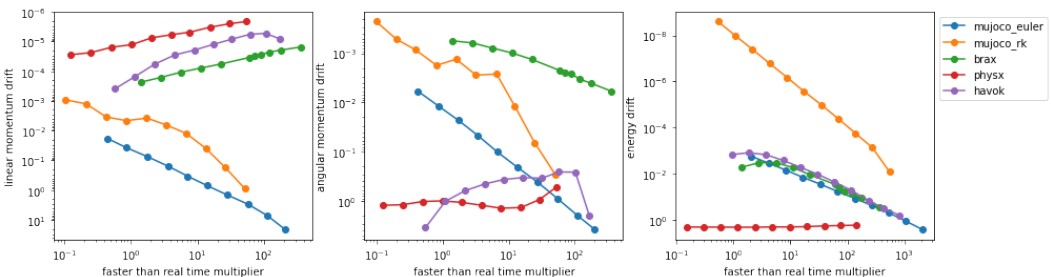

Figure 5: Linear momentum (left), angular momentum (middle), and energy (right) non-conservation scaling for Brax as well as several other engines. Non-Brax data was adapted with permission from the authors of [42] and plotted here for comparison. Following Erez et al., in the momentum conservation scene we disabled damping, collisions, and gravity, and randomly actuated the limbs for 1 second with approximately .5 $Nm$ of torque per actuator per step. For energy, we additionally disabled actuators, gave every body part a random $1m/s$ kick, and measured the energy drift after 1 second of simulation. All measurements averaged over 128 random seeds with single precision floats.

compared to their MuJoCo counterparts. Note that this is not meant to be a claim that we facilitate "higher reward", because comparing different reward functions is somewhat theoretically fraught (though Brax's reward functions are very close to the MuJoCo gym definitions, see Appendix E for more details). We intend only to demonstrate that the progression of reward gain is similar, and that Brax environments achieve qualitatively similar performance over a similar number of learning steps.

Finally, we consider the simulation quality of our engine by how it performs in the "astronaut" diagnostic introduced by [42]—a modified version of the humanoid scene which measures momentum and energy nonconservation as a function of simulation fidelity, depicted in Fig. 5. Qualitatively, Brax achieves competitive linear momentum conservation scaling owing to its maximal cartesian coordinate representation of positions and symplectic integration scheme. Energy conservation performance is in line with Havok and MuJoCo's euler integrator. Brax does exceptionally well at angular momentum conservation, comparatively.

## 7 Leveraging Differentiability

Up until this point, we have focused on the ways in which Brax facilitates fast policy learning by leveraging scale on accelerators. Because Brax was written in JAX, and because we took care to only use differentiable ops in the core physics primitives, it is also possible to learn policies using the gradient of the environment loss directly by employing JAX's *grad* primitive. In this section, we consider a simplified 2-jointed reaching task with identical observation space to OpenAI's MuJoCo Reacher, called brax-reacherangle, but using angle actuation instead of torque actuation (see [43]).

This task is simple enough that it showcases the promise of a differentiable environment without other complexities like contact physics that are known to make optimization difficult even with access to gradients.

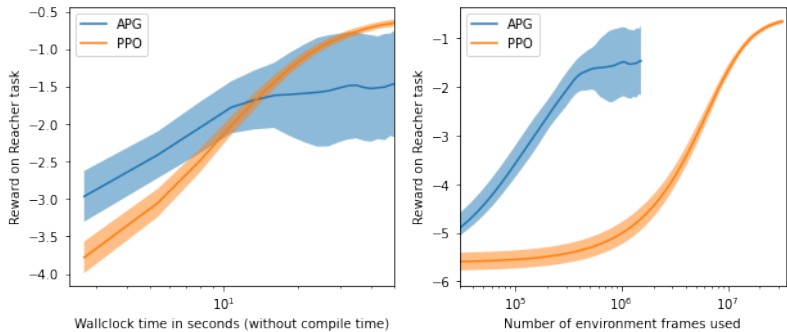

Figure 6: Performance of PPO and APG on the brax-reacherangle task for 30 random seeds. The same data is pictured in both panes, but versus wallclock time (left) and versus number of environment frames used at that point in training (right). Solid lines indicate mean, and shaded regions indicate standard deviation. See App. C for the hyperparameters used for these experiments, and see App. G for a discussion of the hyperparameter search space.

Fig. 6 depicts the performance of PPO vs. APG on this task. While both algorithms solve the environment in a similar amount of time (though PPO is still typically faster), APG achieves dramatically better sample complexity, and is able to solve the environment using an order of magnitude fewer environment frames than PPO. This sort of scaling is expected: policy gradient algorithms are well known to require significantly more environment frames to estimate a gradient than simply calculating the gradient directly [9]. Note that because calculating gradients is expensive, PPO can estimate gradients more quickly than APG can directly calculate them.

One prominent feature of these plots is PPO's noticeably lower training variance, compared to APG's. PPO achieves lower variance training by truncating trajectory unrolls to only 50 steps, compared to APG's full 100-step trajectories. This sort of truncation is known to bias the gradients, but with the advantage that it can make optimization easier[44]. We will explore reducing the variance of APG through other optimization methods like truncated backpropagation through time[45] in future work.

# 8 Limitations and Future Work

In this section, we detail several important limitations and frailties of our engine.

## 8.1 Spring Joints

It is well known that physics engines that rely on spring constraints instead of more sophisticated Featherstone-style methods can be brittle and can require careful tuning of damping forces. Practically, these instabilities arise as a small radius of convergence in the integrator, necessitating small integration step sizes. Worse, these instabilities grow as a function of the difference in mass scale present in a problem. While relying on spring constraints has greatly simplified the core primitives of our engine, it does mean that ensuring stability in a new physics scene can require a fair amount of tuning of damping forces, mass and inertia scale, and integration step size.

Additionally, because our systems are essentially large coupled spring-mass configurations, there is more "jitter" in our simulation traces than in a hypothetical corresponding Featherstone simulation. This can be mitigated by increasing the strength of joint spring constraints, but this comes at the cost of a reduced maximum stable integration step size. For the environments in this release, we chose these spring constants so as to maximize simulation speed while still retaining qualitatively smooth simulation, and we will investigate Featherstone methods in future work.

### 8.2 Collisions

Inspired by the Tiny Differentiable Simulator[16], we use velocity-level collision updates with Baumgarte stabilization for all of our collision primitives. We did experiment with fully springy, impulsive collisions, but found the motion quality and stability to suffer. Because of this choice, we inherit the known tuning requirements and intrinsic non-physicality of these methods[46]. We experimented with time-of-impact based collision detection, but, similar to the authors of DiffTaichi[17], we found it provided little accuracy advantage for the complexity penalty it added to the codebase.

Additionally, we currently only use the quadratically-scaling, naive collision detection for any colliders included in a scene. Typical physics-based sequential decision problems don't involve enough colliders for this to be a significant bottleneck, given that we can still easily parallelize over all collision primitives in a scene without straining modern accelerator memory buffers, but we imagine this will become more strained over time as tasks grow in complexity. We leave more advanced collision physics, e.g. LCP-based solvers, and more efficient collision pruning to a future release.

### 8.3 Jitting, JAX, and XLA

While we tout our ability to compile pythonic physics environments and learning algorithms side-by-side to XLA as a strong comparative advantage that our library inherits from JAX, this does not come without any development friction. Of most salience for end-users of Brax, JIT compilation times can sometimes approach or exceed the training time for complicated environments (i.e., compilation can take minutes). We iterated extensively on the core design patterns of Brax to ameliorate this, and in some cases, collaborated directly with the JAX development team to adjust XLA compilation heuristics on TPU to improve compilation speed and performance. Ultimately, compilation time remains a small bottleneck, particularly for learning algorithms that leverage differentiability.

### 8.4 Algorithms

This work presents results for our PPO and SAC implementations. While we include APG and ES in this release, they have not been as thoroughly tested, nor have we performed as many hyperparameter explorations with them. We leave it to future work to fully leverage the differentiability of our engine.

### 8.5 Social Impacts

Producing another version of what practitioners commonly use almost definitionally further complicates the landscape of existing benchmarks, but we hope that the development velocity unlocked by our library more than makes up for this extra friction. At the same time, the democratizing effect of releasing an engine that can solve control problems quickly can be double edged: the difference between a piece of democratizing technology and a weapon depends entirely on who is wielding it. Mastery over the control of robots represents a society-transforming opportunity, thus we hope our engine only helps to improve and accelerate the equitable automation of our future.

There remains a chance, however, that by releasing a significantly faster engine, we inadvertently dramatically increase the compute spent on reinforcement learning problems, in much the same way building a new highway in a city can counter-intuitively *increase* traffic[47]. At least for our own energy expenditure, the experiments we performed were done in datacenters that are on track to be fully renewably sourced by 2030[48].

## Acknowledgments and Disclosure of Funding

The authors thank Erwin Coumans for invaluable advice on the subtle implementation details of physics engines, Blake Hechtman and James Bradbury for answering the authors numerous questions and providing optimization help with JAX and XLA, Luke Metz and Shane Gu for stimulating feedback and helpful discussions throughout the development of this project, and Yuval Tassa for exceptional feedback on an early draft of this manuscript. The authors further thank Vijay Sundaram, Wright Bagwell, Matthew Leffler, Gavin Dodd, Brad Mckee, and Logan Olson for helping to incubate this project.

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
