# OpenReview forum: "Brax - A Differentiable Physics Engine for Large Scale Rigid Body Simulation"
_NeurIPS.cc/2021/Track/Datasets_and_Benchmarks/Round1 — NeurIPS 2021 Datasets and Benchmarks Track (Round 1)_

### Official Review · Reviewer_Budo · 2021-07-01
**A useful codebase/infrastructure, but not clearly a good academic paper**

**Rating:** 6
**Confidence:** 4

**Strengths:**

1) the code/infrastructure will be very useful and important in advancing the RL research since it provides faster training speed, more differentiability, and re-implementations of many popular RL environments and algorithms.
2) from the experimental results, it is clear to see the benefits in terms of training speed, correctness, and accelerator parallelism.
3) the paper provides many implementation/usage details, extensive APIs/examples, and releases a well-organized/well-documented/informative codebase on Github.

**Weaknesses:**

This paper lacks important comparisons to previous infrastructures and more experimental supports for fully proving improvement in terms of the differentiability of the system.
1) In the last sentence of the first paragraph of Sec. 2, the authors mentioned that there are "many of engine backends are differentiable[15–21, 13]." I think in-depth comparisons (both qualitatively and quantitatively) to these previous works in terms of differentiability are necessary for the publication of this paper.  I don't see much discussion on this from the current submission.
2) In the last sentence of Sec. 6.1, the authors "defer analysis of ES and DTO to future work." Firstly, it would be good to include these experiments in the current submission, instead of future submission. Secondly and more importantly, the experiment DTO is crucial since it is the only experiment out of the four that involves the "differentiability" of the engine, which is one of the selling points of Brax. In the current submission, for now, I don't see any quantitative results on evaluating the "differentiability" of the engine.

Also, the paper writing is more like API documentation for developers, rather than a research paper, as it contains too many implementation details, such as the classes, functions, etc. Some terms are not even clearly explained, for example, what is QP? Furthermore, since Brax is implemented using JAX, it is not clear what implementations are new in Brax and what are the contributions in the previous JAX work.


**Additional Feedback:**

no

Post-rebuttal: I would like to thank the authors for their responses to my questions. I've raised up my rating to 6 since I feel that this work contributes a lot in terms of infrastructure and benchmarking and will clearly be useful to the community. However, more comparisons to similar differential engines [15–21, 13] are still not thorough enough, to my best judgment, even after the revision. For example, is there any new unique feature that previous engines do not support and Brax brings us? Also, can you compare to any of these differential engines for the "task benchmark over differentiability", rather than just comparing to a non-differentiable baseline? Overall, I lean towards accepting this submission given the other reviewers' ratings and the potential contributions of Brax.

**Clarity:**

The paper writing is more like API documentation for developers, rather than a research paper. Many API details are provided, but many important points for a good research paper, e.g. comparisons to previous works, validations of design choices, proof of the benefits of the new systems, are lacking.

**Correctness:**

The claims are mostly correct.
This is an infrastructure paper.
The experiments, in my opinion, do not fully prove the usefulness of the "differentiability" of the engine.

**Documentation:**

The documentation for this infrastructure is good. Licensing/maintenance should be good.

**Relation To Prior Work:**

Not very clearly. For example, how do you compare to [15–21, 13]?

**Summary And Contributions:**

This paper proposes Brax -- a differentiable physics engine, and a set of RL environment and algorithm re-implementation. This work contributes a codebase/infrastructure that increases the training speed of RL algorithms by leveraging computation/accelerator parallelism and provides differentiability for large-scale rigid body simulation. The code is implemented based on JAX and FLAX. The paper provides re-implementation of many popular gym-like RL environments (such as ant, fetch, grasp, half cheetah, and humanoid) and four algorithms (PPO, SAC, ES, and direct policy optimization). Much detailed documentation for code designs and APIs is provided.

---

### Official Review · Reviewer_LQJ8 · 2021-07-04
**Everything about reinforcement learning should run on accelerators**

**Rating:** 7
**Confidence:** 3

**Strengths:**

(1) The work will be very very useful to researchers across the world. The increased speed of learning and the availability of colab notebooks will be highly beneficial to the speeding up how quickly researchers can test new RL algorithms.
(2) The intuition that all the data generation and processing can be done accelerators is a good one - I hope more companies and institutions learn from BRAX so that this approach is soon the defacto standard.

**Weaknesses:**

(1) I had a hard time understanding section 4 , the physics loop. I think the section is presented at too high a level and could benefit from being broadened.
(2) The figures in appendix D do not really convey any information to me without a legend of some kind. Consider eliminating 90% of the curves, pointing the reader to a location where they may find the rest of the curves and adding a legend.
(3) The paper doesn't read like an academic paper. It reads more like a very high level description of what is bound to a useful software suite for researchers.

**Additional Feedback:**

None

**Clarity:**

Most of the paper is well written. The physics loop is presented at too abstract a level - readers may appreciate a more in depth explanation - however I can see this information being added to official documentation available online.

**Correctness:**

As far as I can tell, the claims are correct. The library appears important and easy to use.

**Documentation:**

Yes, the google colab notebooks are well written and easy to follow. I think this work will garner a lot attention when released.

**Ethics:**

The authors final section is worth reading. I do not have anything to add to their thoughts. I appreciate that the authors took this question seriously and answered it in an intellectually honest capacity.

**Relation To Prior Work:**

Related work is mentioned in the motivation section. The work differs from other works in its use of accelerators for simulation. I think the contributions are clear and important and markedly different from the mentioned works.

**Summary And Contributions:**

The authors introduce BRAX, an open source library for rigid body simulation. Standard approaches in RL run simulations on multi-threaded CPUs requiring for data to transferred to accelerators prior to learning. This can form a significant bottleneck and slows down learning immensely. In BRAX all the processing regarding environments and learning occur on accelerators leading to massive speedups in training and testing times. The authors open source their work, provide google colab notebooks to get the uninitiated started quickly and take a step towards massively changing the reinforcement learning landscape by speeding up leaning on Mujoco like environments by 100-1000x.

---

### Official Review · Reviewer_v23e · 2021-07-05
**Review for Brax - A Differentiable Physics Engine for Large Scale Rigid Body Simulation**

**Rating:** 7
**Confidence:** 3
**Correctness:** No obvious incorrectness is noticed.
**Clarity:** Yes

**Strengths:**

- Differentiable physics engine provides a strong alternative for solving robotics tasks other than RL;
- The proposed engine is well optimized to run much faster in terms of environment simulation and algorithm training;
- The proposed engine is well compared to previous engine to validate its correctness;
- The paper is clearly written;

**Weaknesses:**

- I would see the differentiablity of the engine to be one of its main advantages over existing engines like pybullet and mujoco. However, the experiments of DPO that leverages this features are not extensively conducted and analyzed. It would be interesting to see more experiments on this
- Although some discussion is already in the paper, it would be good to see some concrete limitations & failure cases of the proposed engine. For example, how many rigid bodies can the engine simulate simutaneoiusly (the title says large-scale, but there is no experiment on large-scale rigid body simulations)? How accurate is the collision with different geometires of objects for the manipulation tasks?  It would be very important for the users to know such limitations.

**Additional Feedback:**

No

**Documentation:**

The documentation is sufficient.

**Ethics:**

The authors already give a good discussion on the ethical concerns.

**Relation To Prior Work:**

Yes, it is discussed clearly.

**Summary And Contributions:**

This paper proposes a new physics engine for regid body simulation that 1) is differentiable; 2) can run environment simulation and RL algorithm optimization on the same accelerator core without data marshelling or transfering.
It re-implements three mujoco locomotion tasks and two additional manipulation & locomotion tasks to demonstrate the correctness and utility of the engine. It further benchmarks 4 algorithms on the proposed task and compares the engine with the traditional mujoco physics engine to validate it.
The proposed differentiable physics engine, with further development, can be very useful for tackling robotics control tasks like locomotion and manipulation.

---

### Author Response · Authors · 2021-07-12
**Updated revision with summary of changes**

We again thank the reviewers for their comments and suggestions.  We would like to provide a short summary of changes that we have made as a result of reviewer comments, as promised in our responses to individual reviewers in the above comment threads.  All of these changes are now reflected in the most recently uploaded manuscript.

1. We have renamed DTO (DPO in some places…) to APG (Analytic Policy Gradients) under suggestion from an internal reviewer.

2. As requested by Reviewers v23e and Budo, we have added a new Section (Sec. 7 - Leveraging Differentiability) which concerns performance of APG in our engine.  In this section, we provide results on a new task we have added to our repo reminiscent of OpenAI's two-dimensional Reacher task.  We show that by leveraging the gradient of the task loss directly (instead of only estimating it as in PPO), we can learn policies using an order of magnitude fewer environment frames than via PPO on the same task.  Additionally, we have added a new supplemental Appendix G which discusses the hyperparameter sweeps we performed for analyzing this sample efficiency.

3. As requested by Reviewer v23e, we have clarified the caption of Figure 2 to better represent the performance of Brax with respect to large numbers of rigid bodies, and we have added a new supplemental Appendix F which concerns failure cases---i.e., how many rigid bodies it takes before Brax simulations fail on accelerators.

4.  As requested by Reviewer LQJ8 (and to Reviewer Budo to a degree), we have reworked The Physics Loop section (now Section 3---the old Section 3 was moved into the main text elsewhere, as it was just a pointer to our colab notebooks), and clarified how the QP state is transformed in the course of the physics step.

5. As requested by Reviewer LQJ8, we have reworked the figures in Appendix D by reducing the total number of curves plotted and by adding descriptive text and labels that indicate how to find the hyperparameters used to generate each training curve.

6. As requested by Reviewer Budo, we have added a new paragraph to Subsection 6.2 Engine Comparisons, and a new Figure 5 analyzing the performance of Brax on the Astronaut benchmark task introduced by [1] (a variant of the Humanoid task).  This task measures energy nonconservation, and momentum/angular momentum nonconservation as a function of simulation fidelity.

7. Both Reviewers Budo and LQJ8 commented that the manuscript read too much like documentation. With the addition of the extra experiments, along with several editing passes, we have made an attempt to reduce this, though the primary contribution of this work is a new software suite, so some amount of this feel is unavoidable.

8.  Finally, we have done a large number of small tweaks to language, spacing, and content so as to fit the manuscript into the allowed space with the addition of the new figures.

We would be happy to respond to further reviewer comments either in this thread, or in the threads above, as all of the changes promised in those threads have been completed.

All three reviewers have commented positively about the speed, usefulness, and clarity of Brax.  We hope that this suite of changes has addressed the vast majority of Reviewer concerns, and look forward to your final evaluation.


References:
[1] Erez, Tom, Yuval Tassa, and Emanuel Todorov. "Simulation tools for model-based robotics: Comparison of bullet, havok, mujoco, ode and physx." 2015 IEEE international conference on robotics and automation (ICRA). IEEE, 2015.

---

### Decision · Program_Chairs · 2021-07-26

**Decision:**

Accept

**Comment:**

The paper presents a differentiable physics engine in implemented in JAX that can take advantage of modern hardware accelerators. All reviewers saw the benefit of the proposed physics engine and agreed that it would be very useful to practitioners. Initial reviews raised some concerns about the paper, most importantly that comparisons and discussion of other physics engines were lacking, that none of the experiments showcased the benefits of differentiability, and that failure cases were not discussed. Reviewers also felt that the initial draft read more like API documentation than a scientific paper. The author response and revised paper was able to address these concerns, and in the end all reviewers thought that the paper should be accepted. Congratulations on having your paper accepted to the NeurIPS 2021 Datasets and Benchmarks Track! The authors have already taken much of the feedback from reviewers into account in their revised draft, but the authors are encouraged to use the additional space in the camera-ready version of the paper to continue refining the language and presentation in light of the new additions suggested by reviewers.